# HUMAN ALIGNED COMPRESSION FOR ADVERSARIAL ROBUSTNESS

## ABSTRACT

Adversarial attacks on image models threaten system robustness by introducing imperceptible perturbations that cause incorrect predictions. We investigate human-aligned learned lossy compression as a defense mechanism, comparing two learned models (HiFiC and ELIC) against traditional JPEG across various quality levels. Our experiments on ImageNet subsets demonstrate that learned compression methods outperform JPEG, particularly for Vision Transformer architectures, by preserving semantically meaningful content while removing adversarial noise. Even in white-box settings where attackers can access the defense, these methods maintain substantial effectiveness. We also show that sequential compression—applying rounds of compression/decompression—significantly enhances defense efficacy while maintaining classification performance. Our findings reveal that human-aligned compression provides an effective, computationally efficient defense that protects the image features most relevant to human and machine understanding.

## 1 INTRODUCTION

Vision models have made significant improvements in recent years, achieving remarkable success in tasks like image classification He et al. (2015), object detection Szegedy et al. (2013); Zou et al. (2023) and medical imaging Shen et al. (2017). However, vision models remain highly vulnerable to adversarial attacks despite these advancements. Adversarial attacks are carefully crafted perturbations added to input images, which are often imperceptible to the human eye but can cause deep learning models to make incorrect predictions Szegedy et al. (2014); Madry et al. (2019). These attacks pose a serious threat to applications that rely on the reliability of vision models, such as autonomous driving, healthcare diagnostics, and surveillance systems.

One potential approach to defending against these adversarial attacks is to remove the small perturbations introduced by the attacker. Eliminating these imperceptible changes makes it possible to prevent the model from being misled. Traditional methods, such as blurring or adding random noise, can be effective at removing perturbations; however, they have significant drawbacks. While these methods reduce the adversarial impact, they distort the image, changing its distribution in a way that may cause the image to be "out of distribution" for the classifier. Moreover, they do not solely remove adversarial perturbations but also information that could be important for the task. This results in a loss of critical information, degrading model performance, and making the system less reliable.

Ilyas et al. (2019) showed in their seminal work that image classifiers learn to use non-robust features for image classification that the adversarial examples exploit. In the area of image compression, researchers showed that human-perception-aligned learned lossy compression models can yield high compression ratios while producing images that humans prefer, for instance, over JPEG compressed images. These techniques aim to preserve an image's most important features according to human perception while discarding less significant details. By doing so, they can remove adversarial perturbations without altering the underlying image distribution and without losing task-relevant information. Since the images remain in distribution for the classifier, the model can continue to perform effectively while being protected from adversarial attacks. Dziugaite et al. (2016) has shown that JPEG compression could be a viable defence, however, as later shown, only if the attacker does not include the JPEG compression in the attack Shin & Song (2017).

This work explores the potential of learned image compression methods to defend against adversarial attacks. We compare these methods to a traditional technique, JPEG, and evaluate their effectiveness

in removing adversarial perturbations while preserving the integrity of the image distribution. Our findings show that human-perception aligned compression offers a promising strategy for defending vision models against adversarial attacks, without sacrificing classification accuracy. This approach contributes to developing more efficient and robust defense mechanisms, fostering the creation of more secure and reliable vision systems.

## 2 RELATED WORK

### 2.1 ADVERSARIAL ATTACKS

Adversarial examples have become a critical concern in machine learning, particularly regarding the robustness and security of deep learning models. These inputs are intentionally crafted to deceive models into making incorrect predictions. Szegedy et al. (2014) first demonstrated that small, imperceptible perturbations could cause neural networks to misclassify images with high confidence.

Kurakin et al. (2017) extended the study of adversarial attacks to the physical world, showing that printed images with adversarial perturbations could still deceive classifiers when captured through a camera, underscoring the real-world implications of adversarial attacks beyond digital environments.

In response to these challenges, various defense mechanisms have been proposed. Madry et al. (2019) introduced Projected Gradient Decent (PGD) attacks along with adversarial training, a technique in which models are trained on adversarial examples to improve their robustness. Papernot et al. (2016) proposed defensive distillation, leveraging knowledge distillation Hinton et al. (2015)—a technique that compresses an ensemble of models into a smaller model—to enhance resistance against adversarial attacks. Another approach involves preprocessing inputs to remove adversarial perturbations before classification, which can be achieved using image compression techniques Dziugaite et al. (2016). Despite these efforts, achieving a comprehensive defense against adversarial attacks remains an open problem.

### 2.2 IMAGE COMPRESSION AS ADVERSARIAL DEFENSE

Dziugaite et al. (2016) demonstrated that JPEG compression can weaken adversarial attacks by removing small perturbations. The perturbations often vanish by compressing and decompressing a potentially manipulated image, reducing the attack's effectiveness. Even with a high quality factor of 75, JPEG compression enhanced model robustness.

However, Shin & Song (2017) demonstrated that the defensive effectiveness of JPEG compression can be significantly diminished by leveraging a differentiable approximation of the algorithm. By propagating gradient information through the model *and* the JPEG compression process, adversarial attacks can generate perturbations that persist even after the compression and decompression steps.

### 2.3 LEARNED LOSSY IMAGE COMPRESSION

Recent advancements in image compression have moved beyond traditional methods like JPEG, which use linear transformations, to learned techniques that replace the Discrete Cosine Transform (DCT) with nonlinear transformations Liu et al. (2023); He et al. (2021); Galteri et al. (2019); Agustsson et al. (2019); Minnen et al. (2018). Variational Autoencoder (VAE)-based models and Generative Adversarial Networks (GANs) Mentzer et al. (2020) help minimize compression artifacts, producing realistic images even at ultra-low bitrates.

HiFiC Mentzer et al. (2020) leverages GANs to achieve visually appealing reconstructions while preserving perceptually significant information. A user study demonstrated HiFiC's superiority in reconstruction quality over other methods, even at half the bits per pixel.

ELIC He et al. (2022) optimizes image compression for both speed and efficiency. It outperforms previous learned methods, such as Minnen et al. (2018); Cheng et al. (2020), especially at low bitrates.

At low bitrates, learned compression methods outperform JPEG by preserving perceptual quality and reducing visual artifacts, all while maintaining the original image distribution. Thus, using learned compression as preprocessing for neural network classifiers can enhance robustness by removing small perturbations while preserving the image distribution.

## 3 METHODS

Our experiments' objective was to evaluate the effectiveness of image compression as a defense mechanism against adversarial attacks. We also investigated the impact of varying compression quality levels and the effects of applying multiple compression steps sequentially.

We implemented our defenses as an additional preprocessing step applied to the (perturbed) image before it was fed into the classifier. Since lossy image compression inherently removes specific details, it is expected to eliminate some of the adversarial perturbations, thereby reducing the effectiveness of the attack. We then generated adversarial perturbations for the dataset and assessed the classifier's accuracy on the perturbed images, comparing it to the baseline accuracy before the attack.

### 3.1 DEFENSES

The compression methods used as defenses were JPEG, HiFiC, and ELIC. JPEG was selected because it is one of the most widely used compression algorithms and has been previously explored as a defense mechanism against adversarial attacks Dziugaite et al. (2016); Shin & Song (2017). Thus, JPEG serves as a baseline for comparison with the other compression techniques. To implement JPEG compression and decompression, we used the differentiable approximation provided by Kornia Riba et al. (2020).

HiFiC and ELIC are learned compression methods that utilize different architectures to achieve high image quality at low bitrates (Table 1, Mentzer et al. (2020); He et al. (2022)). PyTorch

Table 1: Bits per pixel (BPP) measurements for different compression methods and quality settings, computed on 100 224×224 images from ImageNet. This table enables direct comparison of compression efficiency across JPEG, ELIC, and HiFiC methods at various quality levels.

| JPEG | | ELIC | | HiFiC | |
|---|---|---|---|---|---|
| Quality | BPP | Weights | BPP | Weights | BPP |
| $q = 5.0$ | 0.35 | 0004 | 0.06 | low | 0.15 |
| $q = 10.0$ | 0.48 | 0008 | 0.09 | med | 0.43 |
| $q = 15.0$ | 0.59 | 0016 | 0.14 | high | 0.46 |
| $q = 25.0$ | 0.78 | 0032 | 0.19 | | |
| $q = 35.0$ | 0.94 | 0150 | 0.42 | | |
| $q = 50.0$ | 1.14 | 0450 | 0.69 | | |
| $q = 75.0$ | 1.65 | | | | |
| $q = 95.0$ | 3.80 | | | | |

implementations and weight checkpoints are publicly available for both methods. [1] [2]

For HiFiC and ELIC, we employed differentiable forward functions. The use of differentiable defenses allowed gradient information to propagate through the entire pipeline (including both the model and the defense mechanism). This is expected to reduce the effectiveness of the defense, as it enables the adversarial attack to adapt its perturbations to persist through the compression process. To account for this, we conducted an additional set of experiments using this stronger adaptive attack. In tables throughout this paper, this is annotated with through being true.

### 3.2 ADVERSARIAL ATTACKS

We use these methods to compute adversarial examples:

- Fast gradient sign method (FGSM) Goodfellow et al. (2014).
- Iterative FGSM (iFGSM) Kurakin et al. (2017).
- Projected gradient descent (PGD) Madry et al. (2019).
- Carlini-Wagner attack (CW) Carlini & Wagner (2017).
- DeepFool attack (DeepFool) Moosavi-Dezfooli et al. (2016).

For FGSM, iFGSM and PGD we use different $l_\infty$ norm values (epsilon) to modulate the attack strength, for CW and DeepFool we computed the accuracy for perturbations below a specific $l_2$ norm value. For iFGSM and PGD we used 10 iterations. For all attacks, the torchattacks Kim (2020) implementation was used. For a full list of hyperparameters see Table 3 in the Appendix. When we pass the gradients through the compression model as in Shin & Song (2017), we call it as a "white-box" while if we do not, we call it a "black-box" attack.

---

[1]HiFiC: `https://github.com/Justin-Tan/high-fidelity-generative-compression`.
[2]ELIC: `https://github.com/VincentChandelier/ELiC-ReImplemetation`.

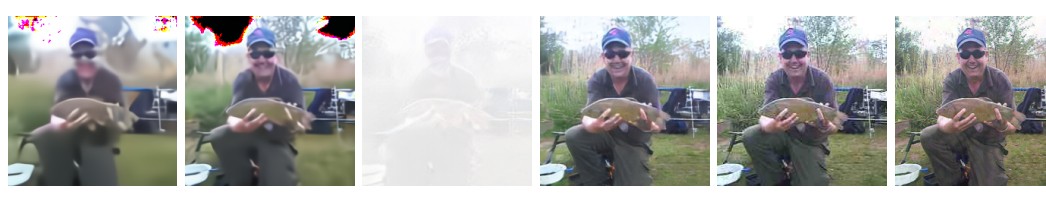

a ELIC low qual-­ity  b ELIC high qual-­ity  c HiFiC low qual-­ity  d HiFiC med. quality  e JPEG quality 25.0  f Original image

Figure 1: Visual comparison of image degradation after three compression/decompression cycles using different compression methods and quality settings. From left to right: (a) ELIC 0004, (b) ELIC quality 0016, (c) HiFiC low, (d) HiFiC medium, (e) JPEG quality 25.0, and (f) the original uncompressed image. Note how learned compression methods (ELIC, HiFiC) exhibit different artifact patterns than traditional JPEG compression.

## 3.3 MODELS AND DATASETS

The experiments used two different base models: ResNet50 He et al. (2015), and a Vision Transformer (ViT), specifically ViT-B/16 Dosovitskiy et al. (2021). Both models were sourced from PyTorch Paszke et al. (2019) and initialized with pretrained ImageNet weights.

For our experiments, we used the validation split of Imagenette, a subset of ImageNet Deng et al. (2009) containing 10 easily classified classes. In later experiments, the full ImageNet test split was utilized.

## 3.4 COMPRESSION STRENGTH

To determine the appropriate compression quality for our experiments, we conducted additional tests for each compression method, comparing different levels to identify an optimal quality setting. Beyond defensive strength, our choice of compression was also influenced by several factors: the impact of the defense on accuracy in the absence of an attack, comparability between different compression methods and related work, and the availability of pretrained weights for learned compression models. Training these models from scratch was beyond the scope of this study.

The parameters influencing compression strength for the different methods were as follows:

- **JPEG:** The quality parameter $q \in [0, 100]$ and controls the quantization strength of the algorithm, with lower values corresponding to greater compression. We compared values $q \in \{5.0, 10.0, 15.0, 25.0, 35.0, 50.0, 75.0, 95.0\}$. Typically, values greater than 70 are considered high quality, while values below 30 result in low-quality images that may appear pixelated and blurry.

- **HiFiC:** Three different sets of pretrained weights were available for HiFiC: **HiFiC$^{\text{low}}$**, **HiFiC$^{\text{med}}$** and **HiFiC$^{\text{high}}$**, which were trained to achieve target bitrates per pixel (BPP) of 0.14, 0.3, and 0.45, respectively.

- **ELIC:** Six different checkpoints were available for ELIC: [0004, 0008, 0016, 0032, 0150, 0450]. These correspond to different values of $\lambda$, the rate-controlling parameter, determining the trade-off between estimated bitrate and image reconstruction distortion (see He et al. (2022) for details).

Additionally, we computed BPP values for images from the ImageNet dataset resized to $224 \times 224$ pixels across different compression methods, cf. Table 1. This allowed a direct comparison of size reduction between techniques.

## 3.5 SEQUENTIAL COMPRESSION

We also conducted experiments on the effectiveness of compressing and decompressing an image multiple times in sequence as a defense. We always propagated the gradients through the defense for these experiments to achieve a stronger attack. The experiments were conducted on Imagenette and ImageNet. For ImageNet, we used 1000 randomly sampled images to reduce computation time.

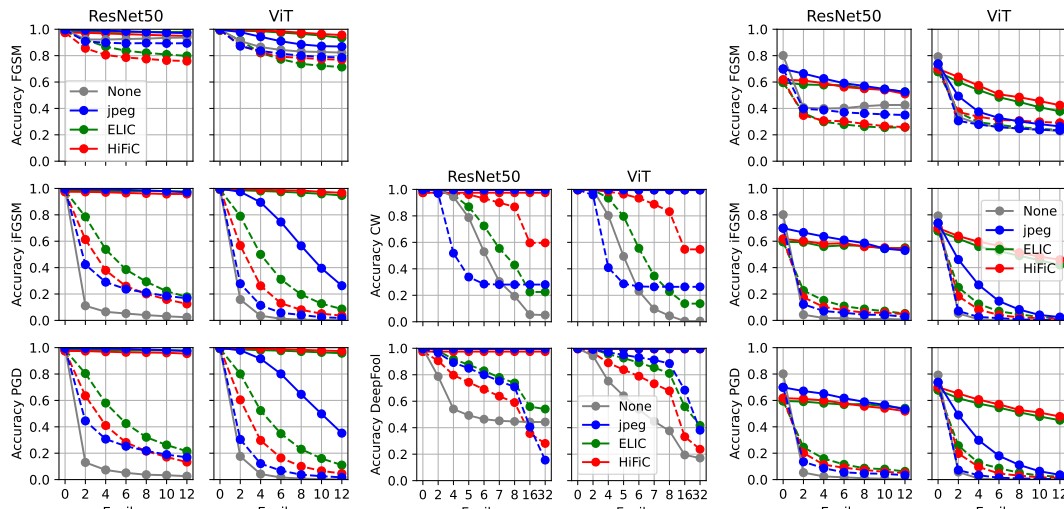

a $L_\infty$ attacks on Imagenette: FGSM (top row), iFGSM (middle row), and PGD (bottom row). Learned compression methods (ELIC, HiFiC) consistently outperform JPEG for the ViT model, particularly under stronger attacks. Epsilon values represent attack strength as $x/255$.

b $L_2$ attacks on Imagenette: Carlini-Wagner (CW, top row) and Deep-Fool (bottom row). Epsilon shows the maximum L2 norm of the perturbations. HiFiC demonstrates superior robustness against CW attacks, especially for the ViT model.

c $L_\infty$ attacks on ImageNet: FGSM (top row), iFGSM (middle row), and PGD (bottom row) on 1000 samples. Results demonstrate that the same defense patterns observed on Imagenette generalize to the more complex ImageNet data, though with lower overall accuracy due to increased task difficulty.

Figure 2: Model accuracy under adversarial attacks for ResNet50 (left) and ViT (right) architectures. Solid lines represent "black-box" attacks (without gradient propagation through the compression), while dashed lines show "white-box" attacks (with gradient propagation through the defense). Results are shown for both Imagenette and ImageNet datasets.

## 4 RESULTS

The main results can be found in Figures 2a and 2b with Tables 4 and 5 in the Appendix giving the exact value. We use quality levels 25.0, low, and 0016 for the compression defenses for JPEG, HiFiC, and ELIC, respectively, unless otherwise stated.

### 4.1 BASELINE RESULTS

Both of the used models achieved a very high accuracy on Imagenette, $\approx 0.998$ for ResNet50 and $\approx 0.999$ for the ViT, as well as a strong resilience against the FGSM attack even without a defense, with both models still achieving accuracies $> 0.8$ at an attack strength of epsilon $\frac{8}{255}$. For iFGSM (cf. Table 2) and PGD the accuracy dropped to $< 0.05$ for ResNet50 and $< 0.01$ for the ViT at this level of attack strength. These baseline experiments also indicate a difference in robustness between the two models used, as the accuracy of the ViT is lower for all baseline experiments.

### 4.2 DEFENSE RESULTS

All three defense methods showed promising results in Figure 2a, showing almost no change in accuracy after all "black-box" attacks when using the ResNet50 model (see Figures 2a and 2b). For "black-box" attacks against ViT, iFGSM and PGD lead to a drop in accuracy at epsilon $\frac{4}{255}$ for JPEG, but still achieve a much higher accuracy than the baseline. The learned compression algorithms show much better performance for ViT, with ELIC and HiFiC achieving an accuracy comparable to before the attack even for high epsilon values.

Adversarial images crafted with the CW and DeepFool attacks were easier to defend against as shown in Figure 2b. This could be attributed to the fact that these attacks find min-

imal perturbations, compared to the other three attacks that find perturbations within the given bounds, which are not necessarily minimal. There are also more hyperparameters to tune for an optimal attack and a longer runtime. These factors lead us to focus less on these attacks and not conduct additional experiments with CW or DeepFool.

Attacking the entire pipeline drastically weakens the effect of all three defenses. There is still improvement over no defense, but the accuracy is not comparable to the experiments where the gradient information was not propagated through the defenses.

These results indicate three major things:

- The ViT used is less robust against adversarial attack.

- The tested learned compressions perform better for ViT.

- Learned compression is a better defence than JPEG.

- Even for learned compression, the effectiveness of compression-based defenses is greatly diminished by creating adversarial images where gradient information was propagated through the defense as seen in Shin & Song (2017).

Table 2: Classification accuracy (%) for different defenses against iFGSM attacks on the Imagenette dataset at varying epsilon values. Results are shown for both ResNet50 and ViT models under black-box (Through=False) and white-box (Through=True) attack scenarios, demonstrating the vulnerability of all defenses to white-box attacks.

|  | Defense | Through | 0 | $\frac{4}{255}$ | $\frac{8}{255}$ | $\frac{12}{255}$ |
|---|---|---|---|---|---|---|
| ResNet50 | None | False | 100 | 6 | 4 | 2 |
|  | JPEG | False | 100 | 99 | 98 | 97 |
|  |  | True | 100 | 28 | 21 | 17 |
|  | ELIC | False | 98 | 98 | 98 | 98 |
|  |  | True | 98 | 54 | 28 | 18 |
|  | HiFiC | False | 98 | 97 | 96 | 96 |
|  |  | True | 98 | 38 | 20 | 12 |
| ViT | None | False | 100 | 4 | 1 | 0 |
|  | JPEG | False | 100 | 90 | 56 | 26 |
|  |  | True | 100 | 11 | 4 | 2 |
|  | ELIC | False | 99 | 98 | 97 | 95 |
|  |  | True | 99 | 50 | 20 | 9 |
|  | HiFiC | False | 100 | 99 | 98 | 97 |
|  |  | True | 100 | 26 | 8 | 4 |

### 4.2.1 IMAGENET RESULTS

After analyzing these results, we decided to repeat some experiments on 1000 random images taken from the ImageNet dataset, to see how the defenses perform on this harder task, with 1000 possible classes instead of 10 and a much lower baseline accuracy (both models achieve an accuracy of about 0.8 on ImageNet). The results of these experiments can be found in Figure 2c. In this experimental setup, all defenses show a larger accuracy decrease without an attack (epsilon= 0). This decrease is larger for ResNet50 than for the ViT. There is also a more noticeable decrease as the epsilon constraint of the attack gets larger for all attacks. On Imagenette the accuracy was almost constant for effective defenses. However, the results on ImageNet generally show the same trends as those discussed before.

### 4.3 COMPUTATIONAL OVERHEAD

We also computed the time it takes the model during inference to show that these defenses are feasible in practice. For this test, we used an Nvidia RTX3090 and ResNet50. We did not include the time the models need to initialize. Without a defense it took 20.8 seconds to classify the 3925 images in the Imagenette dataset, or 5 ms per image. With JPEG as a defense, it took 8 ms per image. Using ELIC and HiFiC it took 14 ms per image. Even when compressing and decompressing the images 5 times in sequence, this only increased to 33 ms for HiFiC and 36 ms for ELIC. These timings show that running these compression algorithms in an ML pipeline is feasible.

### 4.4 QUALITY ABLATION

This section compares different compression strengths for all the compressions used as defenses. Complete tables can be found in the Appendix, see Tables 6 to 8.

### 4.4.1 JPEG

Figure 3a shows the accuracy for different JPEG qualities when attacked with iFGSM. For ResNet50, only the high quality levels (75.0,95.0) were vulnerable to a standard attack. When using the stronger

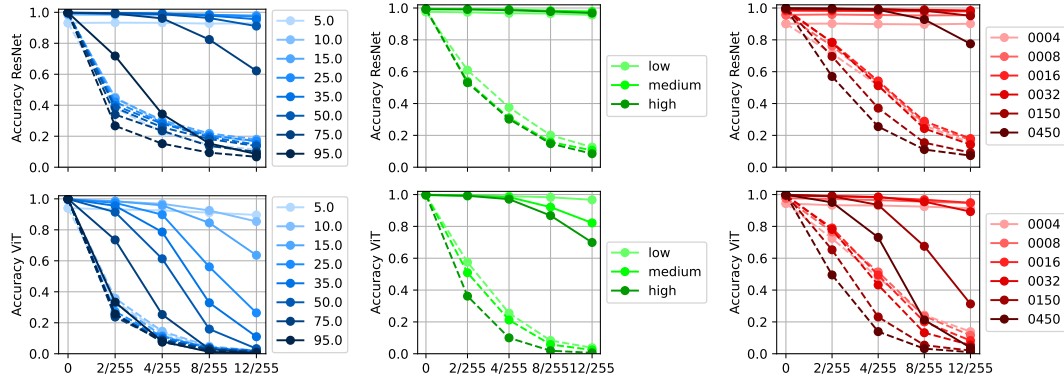

a JPEG quality levels: Lower quality settings provide better defense against strong attacks but reduce clean accuracy, with quality level 25.0 offering the best trade-off.

b HiFiC quality settings (low, medium, high): HiFiC low provides the strongest defense with minimal impact on clean accuracy, particularly for the ViT model.

c ELIC quality parameters (0004 through 0450): ELIC 0016 provides a good balance of defense strength and clean image accuracy, with comparable bitrate to HiFiC low.

Figure 3: Ablation studies comparing model accuracy for different compression quality settings under iFGSM attacks for ResNet50 (top) and ViT (bottom). Dashed lines show results when gradient information was available to the attack ("white-box" setting). All methods demonstrate trade-offs between defense strength and clean accuracy.

attack, which propagates the gradients through the defense, no quality level achieved a high accuracy and therefore a successful defense. For an attack of 8/255 the levels 15.0 and 25.0 achieved the highest accuracy at $\approx 0.2$. In the ViT experiment, a larger spread of results can be observed for "black-box" attacks, with lower quality levels performing better at high epsilon values. JPEG with a quality level 5.0 showed a large decrease in accuracy without any attack. For "white-box" attacks, none of the levels provides much defence. Considering these observations, we decided to use a quality level of 25.0, as this achieves a good performance, does not degrade the baseline accuracy and has been used in previous work by Shin & Song (2017).

### 4.4.2 HiFiC

Figure 3b shows that all the different compression strengths worked well as a defense for ResNet50, but only HiFiC low achieved a high accuracy for the ViT, with the other two qualities showing a decrease in accuracy for epsilon larger than $\frac{4}{255}$. All levels were very vulnerable when the gradient was passed through the defense. Since the lowest quality only decreased baseline performance a little, we used HiFiC low for the main experiments.

### 4.4.3 ELIC

Figure 3c shows the results for ELIC. The higher quality levels show decreased performance for high epsilon values. The lowest quality levels show a decreased accuracy without any attack. There is a larger difference between quality levels when the gradient is propagated through the defense compared to HiFiC. For the main experiment we decided to use ELIC 0016, as it achieved good accuracies without visibly decreasing the baseline performance, and because it is similar in BPP to HiFiC low.

### 4.5 Sequential defense

### 4.5.1 Imagenette

For JPEG, we used quality level 25.0 for the results shown in Figures 4c and 9. For HiFiC and ELIC, there is a tradeoff between a decrease in accuracy without an attack on low qualities and a decrease in effectiveness on high qualities. The baseline accuracy using HiFiC low deteriorates quickly, making it unusable as a sequential defense, see Figure 4a. ELIC 0016 also decreases the baseline accuracy, but

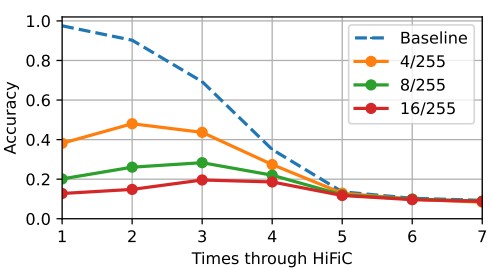

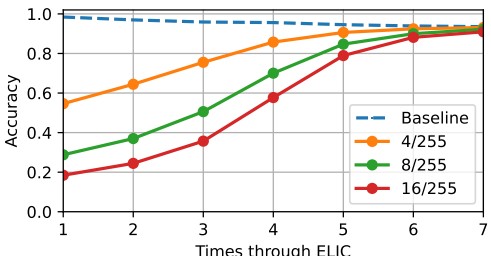

a HiFiC low sequential compression: Baseline accuracy rapidly decreases after multiple iterations, limiting the practical utility of sequential HiFiC defenses.

b ELIC 0016 sequential compression: Accuracy against adversarial examples improves with multiple cycles while maintaining reasonable baseline accuracy, demonstrating better results than HiFiC.

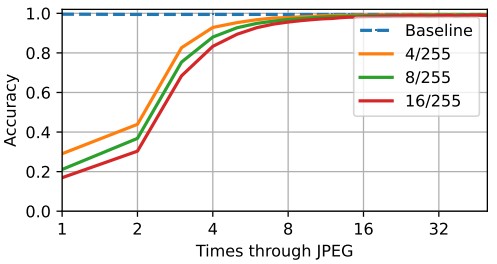

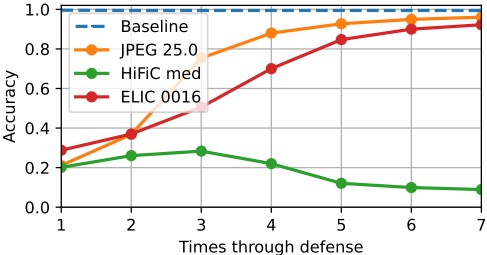

c JPEG defense at different iteration counts: Shows convergence behavior across multiple sequential compression cycles.

d Method comparison for epsilon 8/255: JPEG achieves the fastest convergence toward baseline accuracy with minimal clean accuracy degradation compared to HiFiC and ELIC.

Figure 4: Sequential defense performance analysis against iFGSM attacks on Imagenette using ResNet50. Each point represents N consecutive compression/decompression cycles. JPEG demonstrates superior sequential defense properties with faster convergence and better preservation of clean accuracy compared to learned compression methods.

it was much slower than HiFiC low. This leads to a promising defense that reaches close to baseline accuracy with seven sequential defense iterations, as seen in Figure 4b.

The experiments indicate a clear trend, showing that running an image through a defense multiple times increases its effectiveness for all defenses. JPEG showed the fastest increase, achieving an accuracy of over 0.9 for epsilon $\frac{8}{255}$ after 5 iterations and converging towards the baseline. The decrease in baseline accuracy for JPEG is negligible even after 50 iterations. HiFiC and ELIC also show an increased accuracy for each additional sequential iteration, achieving accuracies of about 0.4 with 7 iterations. Because of the large amount of gradient information, we were unable to compute results for more than 7 iterations of ELIC or HiFiC.

Figure 1 shows how the image quality deteriorates when compressing and decompressing multiple times in sequence. Both of the learned compressions show a stark difference from the original image. ELIC introduces black/red artifacts that take up parts of the image. The lowest quality of HiFiC leads to a much brighter image, which loses most of the color information. JPEG performed best, while it also introduces artifacts and blurs the image a lot, the image still looks similar after multiple passes through the compression.

### 4.5.2 IMAGENET

Experiments in Figures 5a and 5b with sequential JPEG defense on ImageNet yielded similar patterns but with lower overall accuracy, partly due to decreased baseline performance. Lower quality levels reduced clean image accuracy, with this reduction primarily dependent on the quality parameter rather than iteration count, as accuracy remained relatively stable across multiple compression cycles.

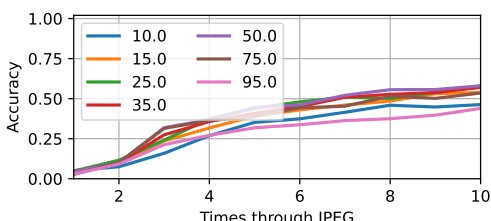

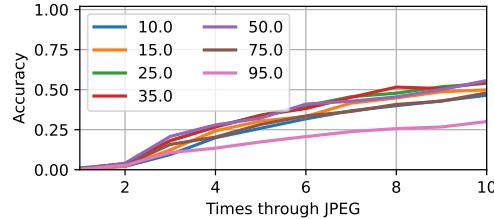

a ResNet50 sequential JPEG defense: Lower quality settings show decreased clean accuracy but provide stronger defenses against adversarial examples.

b ViT sequential JPEG defense: Similar trade-offs between clean accuracy and adversarial defense effectiveness at different quality levels.

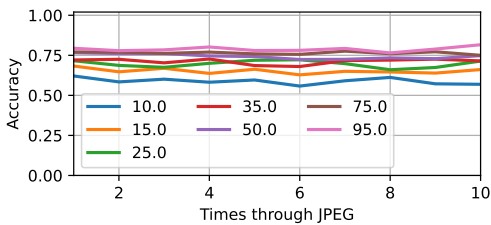

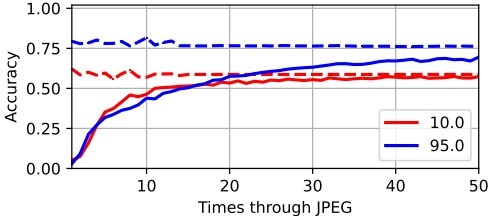

c Baseline accuracy (no attack): Demonstrates how image quality degradation affects classification performance even without adversarial perturbations using ResNet50.

d Extended sequential defense (up to 50 iterations): Quality 10.0 converges faster but to a lower accuracy ceiling, while quality 95.0 continues to improve even after 50 iterations but at a slower rate.

Figure 5: Sequential JPEG defense analysis on ImageNet dataset against iFGSM attacks at various quality levels. Results demonstrate consistent trade-offs between clean accuracy and adversarial robustness across both ResNet50 and ViT architectures, with extended iterations revealing different convergence patterns for different quality settings.

The lines' jaggedness compared to earlier could come from the random sampling. However, even considering the lower baselines, no quality seems to converge towards the baseline with 10 sequential passes. This indicates a deeper sequential defense is needed for this more challenging task.

An experiment with a deeper sequential JPEG defense, see Figure 5d, shows this, as quality 10.0 seems to converge. For quality 95.0 the accuracy still improves even at 50 iterations. The jaggedness of the baseline for low iterations is due to using different subsets of ImageNet. This was changed later in the experiment for more consistent results.

## 5    CONCLUSION

This paper demonstrates that human-aligned learned compression can effectively defend against adversarial attacks. We show that HiFiC and ELIC have advantages over JPEG, as they do not significantly decrease the baseline accuracy of an image classification model even at low BPP. However, we also show the weaknesses of such defenses in a white-box setting, as a gradient can either be directly computed or approximated, decreasing the defense's effectiveness. Sequential compression significantly enhances defense effectiveness, with JPEG showing the most practical balance between robustness and image quality over multiple iterations.

There are some limitations, as we only experiment with gradient-based attacks. In real-world settings, there are more possible ways to attack a model, such as gradient-free attacks Uesato et al. (2018); Engstrom et al. (2019); Gilmer et al. (2018). Further work could include results for these defenses in settings that include these additional threats and combine image compression with other defensive measures to create more robust deep learning models.

While these defenses are weakened in white-box settings, they offer meaningful protection. Future work should explore combining compression-based defenses with other techniques and test against other threats to develop more robust systems that align with human visual perception.

## REPRODUCIBILITY STATEMENT

All code used in our experiments is included in the supplementary material, together with a README file that explains how to set up the environment, run the evaluation scripts, and reproduce the reported results. The data are publicly available through PyTorch's torchvision and Huggingface. For the camera-ready version, we will make the code publicly available on GitHub.

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

# A USAGE OF LLMS

We have used LLMs to polish the writing of this paper and for code generation through chats, cursor, and Claude code. ChatGPT, Claude, Gemini, and Grammarly were employed for spellchecking, refining and condensing text, and reviewing the final draft to catch grammatical errors. Furthermore, ChatGPT, Claude, and Cursor were used to assist with code completion and generate visualizations.

# B TABLES

| Attack | Hyperparameters |
|---|---|
| FGSM | eps=epsilon |
| iFGSM | eps=epsilon,alpha=epsilon/4, steps = 10 |
| PGD | eps=epsilon, alpha=epsilon/4,steps=10,randomstart=True |
| CW | c=1, kappa=0, steps=50, lr=0.01 |
| DeepFool | steps=50, overshoot=0.02 |

Table 3: Hyperparameters used for each adversarial attack method. This table details the specific configuration for the experiments' FGSM, iFGSM, PGD, CW, and DeepFool attacks.

| Attack parameters | | | | | $l_2$ norm value | | | | | | |
|---|---|---|---|---|---|---|---|---|---|---|---|
| Attack | Model | Defense | Through | Baseline | 4 | 5 | 6 | 7 | 8 | 16 | 32 |
| CW | ResNet50 | None | False | 0.998 | 0.944 | 0.787 | 0.529 | 0.304 | 0.193 | 0.054 | 0.051 |
| | | jpeg | False | 0.996 | 0.996 | 0.996 | 0.995 | 0.996 | 0.996 | 0.996 | 0.996 |
| | | | True | 0.996 | 0.518 | 0.338 | 0.285 | 0.281 | 0.281 | 0.281 | 0.281 |
| | | ELIC | False | 0.998 | 0.998 | 0.998 | 0.998 | 0.997 | 0.997 | 0.997 | 0.997 |
| | | | True | 0.998 | 0.958 | 0.869 | 0.724 | 0.554 | 0.429 | 0.225 | 0.225 |
| | | HiFiC | False | 0.975 | 0.975 | 0.975 | 0.975 | 0.975 | 0.975 | 0.975 | 0.975 |
| | | | True | 0.975 | 0.972 | 0.961 | 0.932 | 0.9 | 0.868 | 0.596 | 0.596 |
| | ViT | None | False | 0.999 | 0.803 | 0.494 | 0.231 | 0.097 | 0.045 | 0.005 | 0.005 |
| | | jpeg | False | 0.998 | 0.997 | 0.995 | 0.995 | 0.995 | 0.995 | 0.995 | 0.995 |
| | | | True | 0.998 | 0.409 | 0.287 | 0.267 | 0.264 | 0.264 | 0.264 | 0.264 |
| | | ELIC | False | 0.999 | 0.999 | 0.999 | 0.999 | 0.999 | 0.999 | 0.999 | 0.999 |
| | | | True | 0.999 | 0.934 | 0.796 | 0.555 | 0.346 | 0.228 | 0.137 | 0.137 |
| | | HiFiC | False | 0.995 | 0.995 | 0.995 | 0.995 | 0.995 | 0.995 | 0.995 | 0.995 |
| | | | True | 0.995 | 0.984 | 0.964 | 0.934 | 0.888 | 0.832 | 0.547 | 0.547 |
| DeepFool | ResNet50 | None | False | 0.998 | 0.541 | 0.493 | 0.464 | 0.452 | 0.447 | 0.443 | 0.443 |
| | | jpeg | False | 0.996 | 0.995 | 0.995 | 0.995 | 0.995 | 0.995 | 0.995 | 0.995 |
| | | | True | 0.996 | 0.892 | 0.849 | 0.8 | 0.753 | 0.708 | 0.406 | 0.155 |
| | | ELIC | False | 0.998 | 0.998 | 0.998 | 0.998 | 0.998 | 0.998 | 0.998 | 0.998 |
| | | | True | 0.998 | 0.917 | 0.876 | 0.83 | 0.78 | 0.738 | 0.559 | 0.541 |
| | | HiFiC | False | 0.975 | 0.975 | 0.975 | 0.975 | 0.975 | 0.975 | 0.975 | 0.975 |
| | | | True | 0.975 | 0.797 | 0.743 | 0.69 | 0.638 | 0.59 | 0.356 | 0.281 |
| | ViT | None | False | 0.999 | 0.752 | 0.641 | 0.539 | 0.448 | 0.376 | 0.194 | 0.171 |
| | | jpeg | False | 0.998 | 0.996 | 0.996 | 0.996 | 0.996 | 0.996 | 0.996 | 0.996 |
| | | | True | 0.998 | 0.969 | 0.95 | 0.931 | 0.912 | 0.885 | 0.684 | 0.381 |
| | | ELIC | False | 0.999 | 0.999 | 0.998 | 0.998 | 0.998 | 0.998 | 0.998 | 0.998 |
| | | | True | 0.999 | 0.955 | 0.926 | 0.891 | 0.854 | 0.81 | 0.559 | 0.417 |
| | | HiFiC | False | 0.995 | 0.995 | 0.994 | 0.994 | 0.994 | 0.994 | 0.994 | 0.994 |
| | | | True | 0.995 | 0.89 | 0.838 | 0.789 | 0.732 | 0.678 | 0.332 | 0.237 |

Table 4: Comprehensive evaluation of CW and DeepFool attack effectiveness against different defenses, showing accuracy at various L2 norm constraint values. Results demonstrate that these attacks, which find minimal perturbations, are generally less effective than bounded attacks like iFGSM and PGD.

| Attack parameters | | | | | $l_\infty$ norm value | | | | | |
|---|---|---|---|---|---|---|---|---|---|---|
| Attack | Model | Defense | Through | Baseline | $\frac{2}{255}$ | $\frac{4}{255}$ | $\frac{6}{255}$ | $\frac{8}{255}$ | $\frac{10}{255}$ | $\frac{12}{255}$ |
| FGSM | ResNet50 | None | False | 0.998 | 0.923 | 0.922 | 0.926 | 0.929 | 0.935 | 0.937 |
| | | jpeg | False | 0.996 | 0.994 | 0.99 | 0.984 | 0.979 | 0.974 | 0.971 |
| | | | True | 0.996 | 0.908 | 0.898 | 0.893 | 0.895 | 0.893 | 0.894 |
| | | ELIC | False | 0.983 | 0.983 | 0.979 | 0.981 | 0.98 | 0.979 | 0.98 |
| | | | True | 0.983 | 0.921 | 0.868 | 0.837 | 0.821 | 0.809 | 0.798 |
| | | HiFiC | False | 0.975 | 0.973 | 0.968 | 0.963 | 0.958 | 0.953 | 0.949 |
| | | | True | 0.975 | 0.856 | 0.805 | 0.786 | 0.775 | 0.764 | 0.759 |
| | ViT | None | False | 0.999 | 0.911 | 0.865 | 0.843 | 0.831 | 0.828 | 0.823 |
| | | jpeg | False | 0.998 | 0.978 | 0.943 | 0.909 | 0.885 | 0.872 | 0.869 |
| | | | True | 0.998 | 0.872 | 0.836 | 0.815 | 0.8 | 0.791 | 0.786 |
| | | ELIC | False | 0.992 | 0.989 | 0.985 | 0.975 | 0.965 | 0.952 | 0.935 |
| | | | True | 0.992 | 0.915 | 0.821 | 0.773 | 0.738 | 0.723 | 0.714 |
| | | HiFiC | False | 0.995 | 0.993 | 0.991 | 0.984 | 0.974 | 0.964 | 0.956 |
| | | | True | 0.995 | 0.879 | 0.824 | 0.792 | 0.783 | 0.774 | 0.769 |
| iFGSM | ResNet50 | None | False | 0.998 | 0.111 | 0.065 | 0.052 | 0.04 | 0.031 | 0.024 |
| | | jpeg | False | 0.996 | 0.994 | 0.992 | 0.989 | 0.985 | 0.979 | 0.971 |
| | | | True | 0.996 | 0.423 | 0.29 | 0.237 | 0.21 | 0.185 | 0.168 |
| | | ELIC | False | 0.983 | 0.982 | 0.98 | 0.98 | 0.981 | 0.979 | 0.977 |
| | | | True | 0.983 | 0.785 | 0.538 | 0.385 | 0.293 | 0.221 | 0.178 |
| | | HiFiC | False | 0.975 | 0.974 | 0.971 | 0.966 | 0.962 | 0.958 | 0.958 |
| | | | True | 0.975 | 0.613 | 0.379 | 0.259 | 0.201 | 0.159 | 0.125 |
| | ViT | None | False | 0.999 | 0.158 | 0.035 | 0.013 | 0.006 | 0.004 | 0.002 |
| | | jpeg | False | 0.998 | 0.975 | 0.895 | 0.746 | 0.565 | 0.395 | 0.263 |
| | | | True | 0.998 | 0.278 | 0.114 | 0.058 | 0.041 | 0.024 | 0.018 |
| | | ELIC | False | 0.992 | 0.986 | 0.981 | 0.975 | 0.968 | 0.959 | 0.947 |
| | | | True | 0.992 | 0.79 | 0.5 | 0.311 | 0.196 | 0.127 | 0.086 |
| | | HiFiC | False | 0.995 | 0.993 | 0.991 | 0.986 | 0.984 | 0.974 | 0.967 |
| | | | True | 0.995 | 0.567 | 0.262 | 0.13 | 0.08 | 0.051 | 0.036 |
| PGD | ResNet50 | None | False | 0.998 | 0.13 | 0.073 | 0.051 | 0.037 | 0.033 | 0.026 |
| | | jpeg | False | 0.996 | 0.994 | 0.993 | 0.99 | 0.986 | 0.981 | 0.975 |
| | | | True | 0.996 | 0.445 | 0.307 | 0.253 | 0.218 | 0.189 | 0.169 |
| | | ELIC | False | 0.983 | 0.982 | 0.983 | 0.979 | 0.98 | 0.979 | 0.977 |
| | | | True | 0.983 | 0.804 | 0.58 | 0.425 | 0.321 | 0.262 | 0.215 |
| | | HiFiC | False | 0.975 | 0.974 | 0.969 | 0.965 | 0.963 | 0.961 | 0.955 |
| | | | True | 0.975 | 0.635 | 0.41 | 0.281 | 0.216 | 0.171 | 0.133 |
| | ViT | None | False | 0.999 | 0.176 | 0.042 | 0.016 | 0.008 | 0.003 | 0.003 |
| | | jpeg | False | 0.998 | 0.98 | 0.916 | 0.801 | 0.646 | 0.498 | 0.352 |
| | | | True | 0.998 | 0.304 | 0.121 | 0.068 | 0.037 | 0.026 | 0.017 |
| | | ELIC | False | 0.992 | 0.986 | 0.981 | 0.977 | 0.971 | 0.964 | 0.958 |
| | | | True | 0.992 | 0.801 | 0.523 | 0.348 | 0.23 | 0.16 | 0.11 |
| | | HiFiC | False | 0.995 | 0.993 | 0.992 | 0.988 | 0.984 | 0.978 | 0.974 |
| | | | True | 0.995 | 0.604 | 0.297 | 0.164 | 0.1 | 0.066 | 0.046 |

Table 5: Comprehensive evaluation of FGSM, iFGSM, and PGD attack effectiveness against different defenses, showing accuracy at various L∞ norm constraint values. The table highlights the superior performance of learned compression methods, particularly for the ViT architecture.

## C   SUPPLEMENTING TABLES ON QUALITY ABLATION

| Attack parameters | | | | $l_2$ norm value | | | |
|---|---|---|---|---|---|---|---|
| Model | Quality | Through | Baseline | 2 | 4 | 8 | 12 |
| ResNet50 | 5.0 | False | 0.931 | 0.933 | 0.932 | 0.927 | 0.929 |
| | | True | 0.931 | 0.384 | 0.224 | 0.145 | 0.114 |
| | 10.0 | False | 0.985 | 0.983 | 0.982 | 0.979 | 0.977 |
| | | True | 0.985 | 0.451 | 0.292 | 0.205 | 0.165 |
| | 15.0 | False | 0.991 | 0.99 | 0.988 | 0.984 | 0.979 |
| | | True | 0.991 | 0.444 | 0.303 | 0.218 | 0.18 |
| | 25.0 | False | 0.996 | 0.994 | 0.991 | 0.985 | 0.972 |
| | | True | 0.996 | 0.421 | 0.289 | 0.205 | 0.169 |
| | 35.0 | False | 0.997 | 0.995 | 0.993 | 0.979 | 0.955 |
| | | True | 0.997 | 0.398 | 0.281 | 0.195 | 0.146 |
| | 50.0 | False | 0.997 | 0.995 | 0.988 | 0.963 | 0.911 |
| | | True | 0.997 | 0.384 | 0.258 | 0.182 | 0.137 |
| | 75.0 | False | 0.997 | 0.991 | 0.961 | 0.824 | 0.622 |
| | | True | 0.997 | 0.34 | 0.235 | 0.137 | 0.1 |
| | 95.0 | False | 0.998 | 0.718 | 0.343 | 0.153 | 0.085 |
| | | True | 0.998 | 0.267 | 0.152 | 0.094 | 0.067 |
| ViT | 5.0 | False | 0.94 | 0.932 | 0.925 | 0.911 | 0.896 |
| | | True | 0.94 | 0.337 | 0.135 | 0.042 | 0.016 |
| | 10.0 | False | 0.992 | 0.983 | 0.968 | 0.925 | 0.854 |
| | | True | 0.992 | 0.358 | 0.145 | 0.048 | 0.021 |
| | 15.0 | False | 0.996 | 0.985 | 0.958 | 0.845 | 0.636 |
| | | True | 0.996 | 0.324 | 0.116 | 0.039 | 0.02 |
| | 25.0 | False | 0.998 | 0.974 | 0.898 | 0.561 | 0.264 |
| | | True | 0.998 | 0.272 | 0.111 | 0.037 | 0.018 |
| | 35.0 | False | 0.999 | 0.955 | 0.786 | 0.329 | 0.11 |
| | | True | 0.999 | 0.256 | 0.1 | 0.039 | 0.02 |
| | 50.0 | False | 0.999 | 0.914 | 0.613 | 0.159 | 0.033 |
| | | True | 0.999 | 0.247 | 0.091 | 0.031 | 0.016 |
| | 75.0 | False | 0.999 | 0.735 | 0.253 | 0.03 | 0.006 |
| | | True | 0.999 | 0.237 | 0.087 | 0.028 | 0.012 |
| | 95.0 | False | 0.999 | 0.333 | 0.079 | 0.012 | 0.004 |
| | | True | 0.999 | 0.256 | 0.075 | 0.018 | 0.006 |

Table 6: Ablation studies comparing model accuracy for different quality levels of JPEG compression defense against iFGSM attacks with varying L2 norm constraints. These results informed the selection of optimal quality settings for the main experiments.

| Attack parameters | | | | $l_2$ norm value | | | |
|---|---|---|---|---|---|---|---|
| Model | Quality | Through | Baseline | 2 | 4 | 8 | 12 |
| ResNet50 | low | False | 0.976 | 0.973 | 0.967 | 0.965 | 0.956 |
| | | True | 0.976 | 0.61 | 0.376 | 0.201 | 0.126 |
| | med | False | 0.995 | 0.993 | 0.991 | 0.983 | 0.978 |
| | | True | 0.995 | 0.539 | 0.31 | 0.158 | 0.106 |
| | hi | False | 0.992 | 0.991 | 0.986 | 0.978 | 0.968 |
| | | True | 0.992 | 0.531 | 0.302 | 0.148 | 0.085 |
| ViT | low | False | 0.995 | 0.991 | 0.99 | 0.983 | 0.967 |
| | | True | 0.995 | 0.575 | 0.255 | 0.083 | 0.038 |
| | med | False | 0.999 | 0.995 | 0.984 | 0.921 | 0.822 |
| | | True | 0.999 | 0.509 | 0.212 | 0.058 | 0.024 |
| | hi | False | 0.998 | 0.992 | 0.971 | 0.868 | 0.699 |
| | | True | 0.998 | 0.362 | 0.1 | 0.019 | 0.006 |

Table 7: Ablation studies comparing model accuracy for different quality levels of HiFiC compression defense against iFGSM attacks with varying L2 norm constraints. These results informed the selection of optimal quality settings for the main experiments.

| Attack parameters | | | | $l_2$ norm value | | | |
|---|---|---|---|---|---|---|---|
| Model | Quality | Through | Baseline | 2 | 4 | 8 | 12 |
| ResNet50 | 0004 | False | 0.901 | 0.902 | 0.899 | 0.897 | 0.901 |
| | | True | 0.901 | 0.714 | 0.513 | 0.261 | 0.154 |
| | 0008 | False | 0.957 | 0.959 | 0.956 | 0.953 | 0.955 |
| | | True | 0.956 | 0.761 | 0.534 | 0.275 | 0.17 |
| | 0016 | False | 0.983 | 0.982 | 0.981 | 0.979 | 0.979 |
| | | True | 0.984 | 0.785 | 0.542 | 0.289 | 0.18 |
| | 0032 | False | 0.992 | 0.99 | 0.99 | 0.989 | 0.984 |
| | | True | 0.992 | 0.783 | 0.512 | 0.243 | 0.142 |
| | 0150 | False | 0.998 | 0.997 | 0.994 | 0.984 | 0.951 |
| | | True | 0.998 | 0.696 | 0.371 | 0.154 | 0.092 |
| | 0450 | False | 0.998 | 0.996 | 0.989 | 0.928 | 0.775 |
| | | True | 0.998 | 0.57 | 0.255 | 0.111 | 0.073 |
| ViT | 0004 | False | 0.944 | 0.933 | 0.931 | 0.924 | 0.914 |
| | | True | 0.944 | 0.724 | 0.478 | 0.243 | 0.139 |
| | 0008 | False | 0.977 | 0.972 | 0.965 | 0.955 | 0.947 |
| | | True | 0.977 | 0.765 | 0.514 | 0.236 | 0.116 |
| | 0016 | False | 0.993 | 0.989 | 0.982 | 0.968 | 0.948 |
| | | True | 0.993 | 0.79 | 0.494 | 0.2 | 0.083 |
| | 0032 | False | 0.997 | 0.991 | 0.985 | 0.955 | 0.893 |
| | | True | 0.997 | 0.781 | 0.432 | 0.132 | 0.055 |
| | 0150 | False | 0.999 | 0.989 | 0.935 | 0.675 | 0.313 |
| | | True | 0.999 | 0.653 | 0.231 | 0.054 | 0.02 |
| | 0450 | False | 0.999 | 0.951 | 0.731 | 0.212 | 0.038 |
| | | True | 0.999 | 0.495 | 0.139 | 0.032 | 0.01 |

Table 8: Ablation studies comparing model accuracy for different quality levels of HiFiC compression defense against iFGSM attacks with varying L2 norm constraints. These results informed the selection of optimal quality settings for the main experiments.

| N | Baseline | $\frac{4}{255}$ | $\frac{8}{255}$ | $\frac{16}{255}$ |
|---|---|---|---|---|
| 1.0 | 0.998 | 0.255 | 0.11 | 0.068 |
| 2.0 | 0.998 | 0.32 | 0.134 | 0.086 |
| 3.0 | 0.997 | 0.375 | 0.155 | 0.094 |
| 4.0 | 0.997 | 0.433 | 0.189 | 0.115 |
| 5.0 | 0.998 | 0.501 | 0.231 | 0.145 |
| 6.0 | 0.997 | 0.595 | 0.301 | 0.19 |
| 7.0 | 0.998 | 0.707 | 0.41 | 0.271 |

Table 9: Detailed results for sequential defense using ELIC, showing accuracy after N compression/decompression cycles at different attack strengths.

| N | Baseline | $\frac{4}{255}$ | $\frac{8}{255}$ | $\frac{16}{255}$ |
|---|---|---|---|---|
| 1.0 | 0.995 | 0.312 | 0.162 | 0.101 |
| 2.0 | 0.993 | 0.359 | 0.182 | 0.116 |
| 3.0 | 0.99 | 0.399 | 0.211 | 0.135 |
| 4.0 | 0.989 | 0.444 | 0.245 | 0.153 |
| 5.0 | 0.989 | 0.498 | 0.284 | 0.177 |
| 6.0 | 0.989 | 0.549 | 0.329 | 0.213 |
| 7.0 | 0.987 | 0.602 | 0.385 | 0.253 |

Table 10: Detailed results for sequential defense using HiFiC, showing accuracy after N compression/decompression cycles at different attack strengths.

| N | Baseline | $\frac{4}{255}$ | $\frac{8}{255}$ | $\frac{16}{255}$ |
|---|---|---|---|---|
| 1.0 | 0.996 | 0.29 | 0.211 | 0.169 |
| 2.0 | 0.994 | 0.439 | 0.368 | 0.303 |
| 3.0 | 0.994 | 0.826 | 0.754 | 0.684 |
| 4.0 | 0.994 | 0.929 | 0.88 | 0.833 |
| 5.0 | 0.994 | 0.953 | 0.927 | 0.894 |
| 6.0 | 0.994 | 0.968 | 0.949 | 0.928 |
| 7.0 | 0.994 | 0.976 | 0.961 | 0.946 |
| 8.0 | 0.994 | 0.979 | 0.969 | 0.956 |
| 9.0 | 0.994 | 0.982 | 0.973 | 0.963 |
| 10.0 | 0.994 | 0.984 | 0.978 | 0.968 |
| 11.0 | 0.994 | 0.989 | 0.981 | 0.972 |
| 12.0 | 0.994 | 0.989 | 0.984 | 0.974 |
| 13.0 | 0.994 | 0.991 | 0.984 | 0.978 |
| 14.0 | 0.994 | 0.988 | 0.986 | 0.98 |
| 15.0 | 0.994 | 0.992 | 0.987 | 0.98 |
| 16.0 | 0.994 | 0.991 | 0.988 | 0.983 |
| 17.0 | 0.994 | 0.99 | 0.99 | 0.984 |
| 18.0 | 0.994 | 0.99 | 0.988 | 0.984 |
| 19.0 | 0.994 | 0.993 | 0.988 | 0.985 |
| 20.0 | 0.994 | 0.991 | 0.989 | 0.985 |
| 21.0 | 0.994 | 0.992 | 0.99 | 0.987 |
| 22.0 | 0.994 | 0.991 | 0.989 | 0.987 |
| 23.0 | 0.994 | 0.992 | 0.989 | 0.986 |
| 24.0 | 0.994 | 0.993 | 0.989 | 0.988 |
| 25.0 | 0.994 | 0.992 | 0.99 | 0.988 |
| 26.0 | 0.994 | 0.993 | 0.99 | 0.987 |
| 27.0 | 0.994 | 0.994 | 0.991 | 0.988 |
| 28.0 | 0.994 | 0.992 | 0.99 | 0.988 |
| 29.0 | 0.994 | 0.993 | 0.99 | 0.985 |
| 30.0 | 0.994 | 0.994 | 0.99 | 0.987 |
| 31.0 | 0.994 | 0.992 | 0.99 | 0.987 |
| 32.0 | 0.994 | 0.993 | 0.992 | 0.989 |
| 33.0 | 0.994 | 0.994 | 0.991 | 0.988 |
| 34.0 | 0.994 | 0.993 | 0.991 | 0.989 |
| 35.0 | 0.994 | 0.994 | 0.993 | 0.989 |
| 36.0 | 0.994 | 0.994 | 0.992 | 0.99 |
| 37.0 | 0.994 | 0.994 | 0.99 | 0.986 |
| 38.0 | 0.994 | 0.993 | 0.992 | 0.987 |
| 39.0 | 0.994 | 0.994 | 0.992 | 0.988 |
| 40.0 | 0.994 | 0.993 | 0.992 | 0.989 |
| 41.0 | 0.994 | 0.993 | 0.993 | 0.99 |
| 42.0 | 0.994 | 0.993 | 0.993 | 0.988 |
| 43.0 | 0.994 | 0.993 | 0.991 | 0.991 |
| 44.0 | 0.994 | 0.994 | 0.993 | 0.99 |
| 45.0 | 0.994 | 0.993 | 0.991 | 0.99 |
| 46.0 | 0.994 | 0.994 | 0.993 | 0.991 |
| 47.0 | 0.994 | 0.994 | 0.992 | 0.99 |
| 48.0 | 0.994 | 0.994 | 0.993 | 0.99 |
| 49.0 | 0.994 | 0.994 | 0.992 | 0.989 |
| 50.0 | 0.994 | 0.994 | 0.99 | 0.99 |

Table 11: Detailed results for sequential defense using JPEG show accuracy after N compression/decompression cycles at different attack strengths. JPEG demonstrates superior scaling with iteration count, maintaining high (> 99%) accuracy even after 50 cycles.

# D   GRAPHS

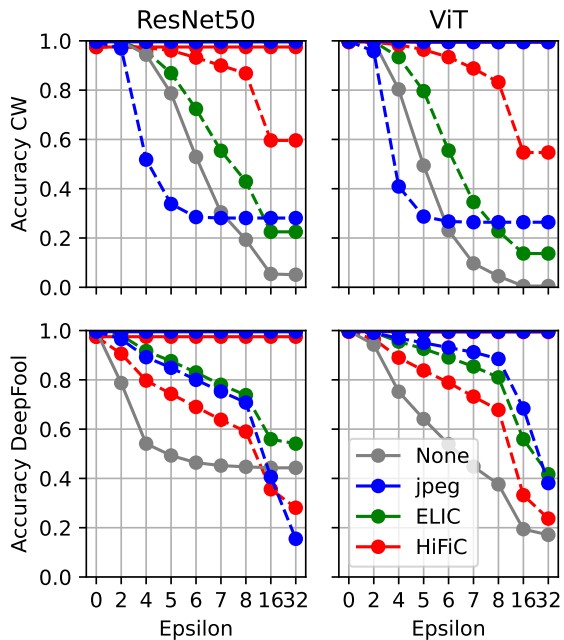

Figure 6: Figure 2b in a larger format

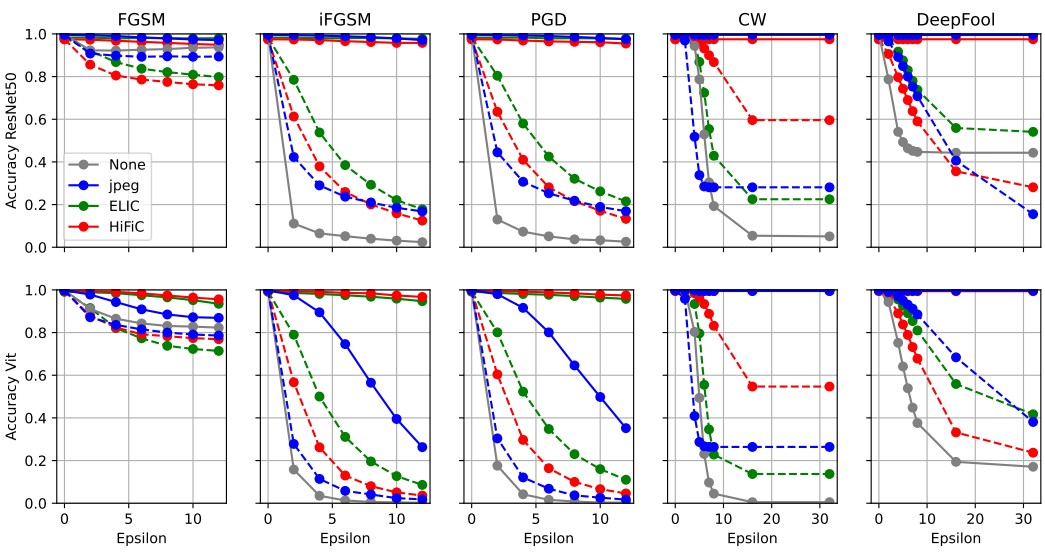

Figure 7: All attacks

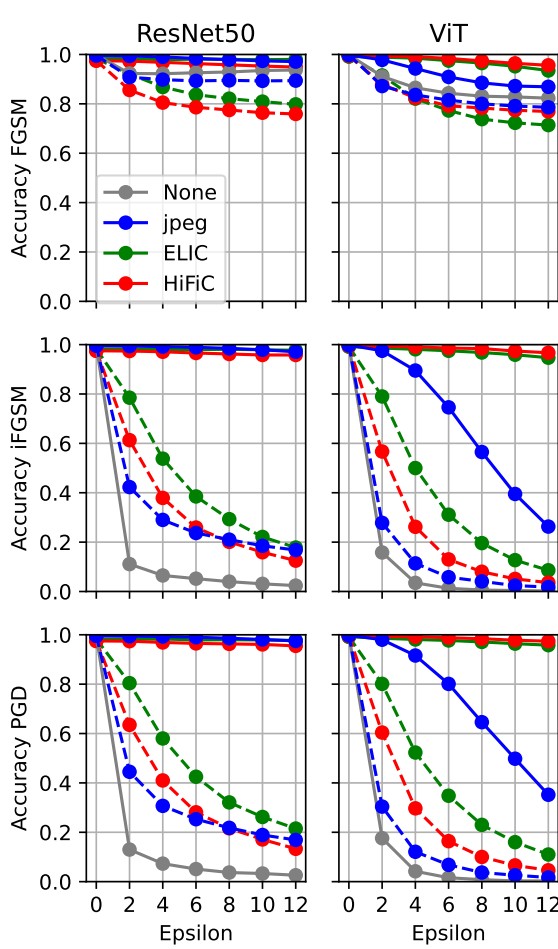

Figure 8: Figure 2a in a larger format

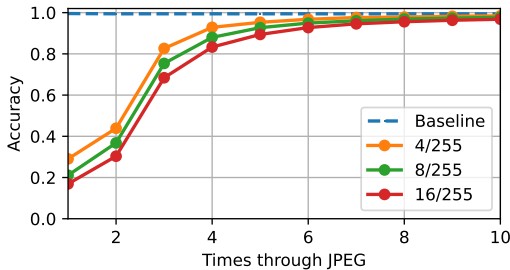

Figure 9: Accuracy of the sequential defense using multiple iterations of JPEG compression and decompression. iFGSM, ResNet50

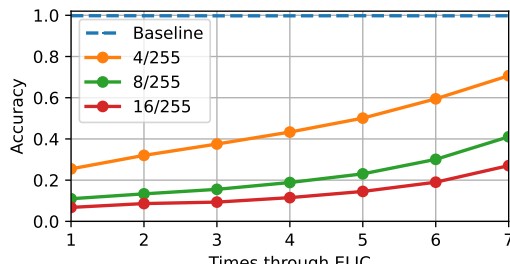

Figure 10: Accuracy of the sequential defense using multiple iterations of ELIC compression and decompression. iFGSM, ResNet50

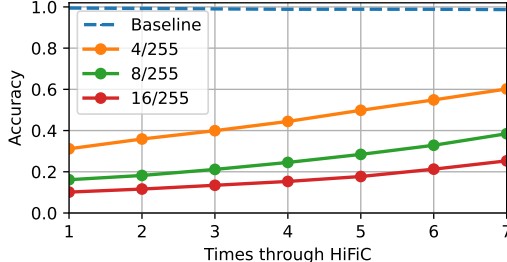

Figure 11: Accuracy of the sequential defense using multiple iterations of HiFiC compression and decompression. iFGSM, ResNet50

