# OpenReview forum: "Human Aligned Compression for Robust Models"
_ICLR.cc/2026/Conference — ICLR 2026 Conference Withdrawn Submission_

### Official Review · Reviewer_W3pH · 2025-10-30

**Soundness:** 2
**Presentation:** 1
**Contribution:** 1
**Rating:** 2
**Confidence:** 4

**Summary:**

This paper investigates the feasibility of using learnable lossy compression models as a defense mechanism against adversarial examples. The experiments demonstrate that HiFiC and ELIC algorithms offer significant advantages over JPEG, showing that even under low BPP (bits per pixel) conditions, they do not drastically degrade the baseline classification accuracy of image models. The paper also highlights the limitations of such defense mechanisms in white-box settings. Furthermore, it is pointed out that multi-round compression techniques can significantly enhance defense effectiveness, with JPEG demonstrating the best balance between robustness and image quality across multiple iterations.

**Strengths:**

1. The experiments are relatively systematic, involving a comparison of the representative learnable compression methods HiFiC and ELIC with the traditional JPEG method. These experiments confirm that the former provides better defense efficacy, offering insights for researchers aiming to improve adversarial robustness through data preprocessing techniques.
2. The discovery of the "multi-round compression" approach, though simple, provides valuable ideas for lightweight defense strategies.

**Weaknesses:**

1. The paper lacks substantial innovation. A similar systematic study was conducted by Dziugaite et al.[1], which already explored this area in depth. This paper merely replaces JPEG with HiFiC/ELIC without offering any significant new contributions. Additionally, there is a lack of theoretical analysis explaining why "human perception-aligned compression" improves model robustness.
2. The scope of the models and datasets used is too narrow. The study only considers two models—ResNet50 and ViT—and primarily utilizes the ImageNet dataset. Other classic models, such as VGG, Inception, and AlexNet, are missing from the analysis.
3. While the paper compares the performance of traditional JPEG and learnable compression methods (HiFiC and ELIC) in defense tasks, as well as the effects of multi-round compression strategies, it lacks a mechanistic explanation. The approach seems more empirical and lacks a convincing theoretical foundation to support its findings.

[1]Dziugaite G K, Ghahramani Z, Roy D M. A study of the effect of jpg compression on adversarial images[J]. arXiv preprint arXiv:1608.00853, 2016.

**Questions:**

1. How is "human perception alignment" specifically defined and measured in the context of this study?
2. I believe the choice of models in the experimental section may be insufficient. Could you explain why only ResNet50 and ViT were selected?
3. What is the underlying mechanism behind the improvement brought by multi-round compression? Can it be explained from an information entropy perspective?
4. In comparing traditional methods with the current learnable compression techniques, are there more innovative methods to address the limitations observed in both?

---

### Official Review · Reviewer_hfuX · 2025-10-30

**Soundness:** 1
**Presentation:** 3
**Contribution:** 2
**Rating:** 2
**Confidence:** 3

**Summary:**

This paper investigates human-aligned learned lossy compression as a defense against adversarial attacks for image models. The authors experimentally demonstrate that learned compressions outperform traditional JPEG and can remain effective even in a challenging white-box setting. Moreover, sequentially applying multiple rounds of compression can further enhance the efficacy of the defense.

**Strengths:**

The authors explored learned image compression models for both single-pass and sequential defense, and conducted a thorough investigation on the relationship between defense efficacy and the number of iterations through the compression module.

**Weaknesses:**

- One major concern is the effectiveness against adaptive attacks. The experiments only considered a simple “white-box” attack where the gradients are backpropagated through the image compression module. This approach may lead to an overestimation of defense efficacy due to potential gradient obfuscation rather than true robustness [1]. The paper would benefit significantly from considering stronger adaptive attacks, which are the recognized standard for evaluating adversarial defenses within the community [2,3].

- The authors claim that the learned image compression defense maintains "substantial effectiveness" even when the adversary has access to the defense. However, this claim appears to conflict with experimental results in later sections, where both ELIC and HiFiC failed to resist iFGSM attacks for ResNet50 and ViT in the white-box setting. Furthermore, the effectiveness of sequentially composed defenses is difficult to ascertain without proper evaluation against adaptive attacks.

- The paper mainly discusses early defense mechanisms such as defensive distillation and JPEG compression (both from 2016). A comprehensive comparison with more recent and advanced defenses (e.g., adversarial purification [4,5]) is absent, limiting the paper's contribution to the current state of research.

- Existing research [6,7] indicates that neural image compression models themselves can be vulnerable to adversarial attacks. Introducing these models as a pre-processing defense module may thus create new attack surfaces, which the current paper does not address.


[1] Athalye, Anish, Nicholas Carlini, and David Wagner. "Obfuscated gradients give a false sense of security: Circumventing defenses to adversarial examples." International conference on machine learning. PMLR, 2018.
[2] Tramer, Florian, et al. "On adaptive attacks to adversarial example defenses." Advances in neural information processing systems 33 (2020): 1633-1645.
[3] Croce, Francesco, and Matthias Hein. "Reliable evaluation of adversarial robustness with an ensemble of diverse parameter-free attacks." International conference on machine learning. PMLR, 2020.
[4] Yoon, Jongmin, Sung Ju Hwang, and Juho Lee. "Adversarial purification with score-based generative models." International Conference on Machine Learning. PMLR, 2021.
[5] Nie, Weili, et al. "Diffusion Models for Adversarial Purification." International Conference on Machine Learning. PMLR, 2022.
[6] Sui, Yang, et al. "Reconstruction distortion of learned image compression with imperceptible perturbations." arXiv preprint arXiv:2306.01125 (2023).
[7] Wu, Chenhao, et al. "On the Adversarial Robustness of Learning-based Image Compression Against Rate-Distortion Attacks." arXiv preprint arXiv:2405.07717 (2024).

**Questions:**

Please refer to the weaknesses listed above.

---

### Official Review · Reviewer_VPbr · 2025-10-31

**Soundness:** 2
**Presentation:** 1
**Contribution:** 1
**Rating:** 2
**Confidence:** 4

**Summary:**

This paper investigates the feasibility of using learnable lossy compression models as a defense mechanism against adversarial examples. The experiments demonstrate that HiFiC and ELIC algorithms offer significant advantages over JPEG, showing that even under low BPP (bits per pixel) conditions, they do not drastically degrade the baseline classification accuracy of image models. The paper also highlights the limitations of such defense mechanisms in white-box settings. Furthermore, it is pointed out that multi-round compression techniques can significantly enhance defense effectiveness, with JPEG demonstrating the best balance between robustness and image quality across multiple iterations.

**Strengths:**

1. The experiments are relatively systematic, involving a comparison of the representative learnable compression methods HiFiC and ELIC with the traditional JPEG method. These experiments confirm that the former provides better defense efficacy, offering insights for researchers aiming to improve adversarial robustness through data preprocessing techniques.
2. The discovery of the "multi-round compression" approach, though simple, provides valuable ideas for lightweight defense strategies.

**Weaknesses:**

1. The paper lacks substantial innovation. A similar systematic study was conducted by Dziugaite et al.[1], which already explored this area in depth. This paper merely replaces JPEG with HiFiC/ELIC without offering any significant new contributions. Additionally, there is a lack of theoretical analysis explaining why "human perception-aligned compression" improves model robustness.
2. The scope of the models and datasets used is too narrow. The study only considers two models—ResNet50 and ViT—and primarily utilizes the ImageNet dataset. Other classic models, such as VGG, Inception, and AlexNet, are missing from the analysis.
3. While the paper compares the performance of traditional JPEG and learnable compression methods (HiFiC and ELIC) in defense tasks, as well as the effects of multi-round compression strategies, it lacks a mechanistic explanation. The approach seems more empirical and lacks a convincing theoretical foundation to support its findings.
[1]Dziugaite G K, Ghahramani Z, Roy D M. A study of the effect of jpg compression on adversarial images[J]. arXiv preprint arXiv:1608.00853, 2016.

**Questions:**

1. How is "human perception alignment" specifically defined and measured in the context of this study?
2. I believe the choice of models in the experimental section may be insufficient. Could you explain why only ResNet50 and ViT were selected?
3. What is the underlying mechanism behind the improvement brought by multi-round compression? Can it be explained from an information entropy perspective?
4. In comparing traditional methods with the current learnable compression techniques, are there more innovative methods to address the limitations observed in both?

---

### Official Review · Reviewer_G1bP · 2025-11-01

**Soundness:** 3
**Presentation:** 3
**Contribution:** 2
**Rating:** 2
**Confidence:** 4

**Summary:**

This manuscinvestigate humanaligned learned lossy compression as a defense mechanism, comparing two learned
models (HiFiC and ELIC) against traditional JPEG across various quality levels.

**Strengths:**

This manuscript systematically compares the impact of different types of image compression codecs on the robustness of image classification models, providing some ideas for experimental settings in this direction of research.

**Weaknesses:**

The technological innovation of the current version is limited:
1) The main contribution of the entire manuscript is the exploratory experiments conducted on image classification tasks using different image compression codecs. There are some new discoveries in the article, but no new solutions are proposed.
2) The title of the manuscript covers a wide range, but the range of evaluation tasks of the experimental setting is limited. The current version is only for image classification tasks and lacks testing for other more complex tasks.
3) The idea of this manuscript is interesting, but it gives the reviewer the impression that it has not been completed yet. The current version is more like a stage baseline testing report rather than a complete academic research paper.

**Questions:**

N/A

---

### Note · Authors · 2025-11-12

I have read and agree with the venue's withdrawal policy on behalf of myself and my co-authors.